# Prescription of Opioid Analgesics for Chronic Non-Cancer Pain in Germany despite Contraindications: Administrative Claims Data Analysis

**DOI:** 10.3390/ijerph21020180

**Published:** 2024-02-05

**Authors:** Anja Niemann, Nils F. Schrader, Christian Speckemeier, Carina Abels, Nikola Blase, Milena Weitzel, Anja Neumann, Cordula Riederer, Joachim Nadstawek, Wolfgang Straßmeir, Jürgen Wasem, Silke Neusser

**Affiliations:** 1Institute for Health Care Management and Research, University Duisburg-Essen, Thea-Leymann-Str. 9, 45127 Essen, Germany; 2DAK-Gesundheit, Nagelsweg 27, 20097 Hamburg, Germany; 3Association of German Doctors and Psychotherapists Practicing in Pain Medicine and Palliative Care (BVSD e.V.), Katharinenstraße 8, 10711 Berlin, Germany

**Keywords:** contraindication, opioid analgesics, long-term prescription, chronic non-cancer pain, guideline, logistic regression, administrative claims data, pain in somatoform disorders, severe mood disorders, primary headaches

## Abstract

In Germany, long-term opioid treatment (L-TOT) for chronic non-tumor pain (CNTP) is discussed as not being performed according to the German guideline on L-TOT for CNTP. In the present analysis, the occurrence and predictors of inappropriate care/overuse in a cohort of German insureds with L-TOT for CNTP by the presence of a contraindication with concurrent opioid analgesic (OA) therapy were investigated. We also analyzed whether prescribing physicians themselves diagnosed a contraindication. The retrospective cohort study was based on administrative claims data from a German statutory health insurance. Eight contraindication groups were defined based on the German guideline. Logistic regressions were performed in order to identify predictors for OA prescriptions despite contraindications. The possible knowledge of the prescribing physician about the contraindication was approximated by analyzing concordant unique physician identification numbers of OA prescriptions and contraindication diagnoses. A total of 113,476 individuals (75% female) with a mean age of 72 years were included. The most common documented contraindications were primary headaches (8.7%), severe mood disorders (7.7%) and pain in somatoform disorders (4.5%). The logistic regressions identified a younger age, longer history of OA therapy, opioid related psychological problems, and outpatient psychosomatic primary care as positive predictors for all contraindication groups.

## 1. Introduction

Opioid analgesics (OAs), a highly effective pain-relieving medication group, are available as one possible option for the treatment of severe chronic non-tumor pain (CNTP) [1]. The German evidence-based guideline [1] provides recommendations for the long-term use of OAs for chronic non-tumor pain (CNTP) and identifies indications and contraindications. For the definition of the long-term use of OAs, different time periods are given in the literature [2]. The frequently used definition for long-term use as a period of OA use ≥90 days is taken as a basis, which is also consistent with the German guideline [1,2]. Moreover, an OA is not recommended as a first-line therapy; the optimization of non-drug and non-OA medication alternatives before initiating OA therapy is recommended [1]. The use of OAs harbors risks in terms of strong side effects, abuse, addiction, and premature death [1,3].

In Germany, the prescription of most OAs is highly regulated. Opioids can only be prescribed on special narcotic prescription forms issued by the Federal Opium Agency [4]. This public authority monitors the entire supply chain of narcotic drugs [5]. Weak opioids, such as codeine, tramadol, and tilidine/naloxone, can be prescribed by physicians via regular prescriptions without restrictions from the German Narcotic Drugs Act [6]. Nevertheless, Germany holds one of the highest prescription rates for OAs, with 879.04 milligrams of morphine equivalent per 1000 inhabitants per day in 2019, ahead of the U.S., surpassed only by Canada and Switzerland [7]. Hence, the question arises about the underlying causes. On the one hand, high prescription rates may result from a reduction in underuse and be favored by an aging and increasingly multimorbid population. On the other hand, they could be an indication of overuse and inappropriate care due to contraindications for long-term OA prescriptions defined in the German guideline [1], among others.

Therefore, the objective of this analysis is to describe the health care situation in Germany regarding evidence of inappropriate care or overuse due to contraindications. Predictors for this issue were investigated. Further, we analyzed to what extent it can be assumed that prescribing physicians were aware of contraindications.

This analysis is part of the Op-US project (Opioid Analgesics—trends in opioid care for chronic non-cancer pain in Germany, Innovation Fund of the Joint Federal Committee of Germany, grant number 01VSF19059). The project aims to investigate the extent to which evidence of overuse, misuse, abuse, or addiction can be identified and attributed to specific subgroups in a cohort of insured persons of a large German statutory health insurance (SHI) provider with the long-term use of OAs. The project methods are described in more detail in the published study protocol [8]. Underlying data for this analysis are administrative claims data, which are secondary data originally collected for billing purposes and can be used in pseudonymized form for research purposes [9].

## 2. Materials and Methods

The retrospective cohort study was based on administrative claims data from the SHI provider “DAK Gesundheit”, which insured 5.7 million persons across Germany in 2018. The cohort was selected during the period 1 January 2018 to 30 June 2019. Each person was followed for two years or until death. Inclusion criteria were defined as insured with long-term OA use, which was specified as prescriptions of OAs in at least two consecutive quarters (selection period). The first and last prescription had to be at least 90 days apart if a person received opioids in only two quarters. Relevant OAs were identified using Anatomical Therapeutic Chemical (ATC) codes, starting with N02A (morphine, hydromorphone, oxycodone, oxycodone and naloxone, fentanyl, buprenorphine, tilidine and naloxone, tramadol, tramadol and paracetamol, tapentadol). Only insured persons who were 18 years or older in the first observation quarter were included. Furthermore, inclusion required continuous insurance coverage from 1 January 2017 to 31 March 2021. Exclusion criteria were a cancer diagnosis in at least one of the two consecutive quarters of the selection period, namely ICD-10-GM (International Statistical Classification of Diseases and Related Health Problems, 10th revision, German Modification) C00-C97, ending with G (secured) or Z (state after) for outpatient care or for inpatient care, as well as evidence of palliative care, namely ICD-10-GM Z51.5; EBM (Uniform Value Scale for outpatient billing) 01425–01426, 03370–03373, 04370–04373, 37300–37320; additional charges ZE60, ZE145, ZE2020-133, ZE2020-134 for inpatient billing; OPS (German procedural classification) 8–982, 8–98e, 8–98h, 1–265.b, 1–773, 1–774. For this analysis, the master data of the insured persons, data from inpatient and outpatient care, and data on drug prescriptions were used [8].

Hereafter, prescribing OAs despite the presence of a contraindication described in the German guideline [1] will be defined as improperly provided health service by physicians (see Table 1), according to a definition of the German expert council on health and care [10]. In the following analysis, it is assumed that there is a subjective need for OA therapy on the part of the patient. If there is an objective need for the therapy, defined by the presence of an indication according to the German guideline [1], the term inappropriate care is used. If there is no indication according to the German guideline [1], it is assumed that there is no objective need for the therapy, which is defined as both overuse and inappropriate care [10].

The aim of the first analysis was to identify inappropriate care and, if applicable, overuse regarding contraindications for OAs, which were coded simultaneously with OA prescriptions. The German guideline [1] lists contraindications to OA therapy for CNTP and provides a basis for the eight defined contraindication groups (see Table 2). Associated ICD-10-GM codes were used to identify contraindications in the study population. It should be noted that the contraindication “chronic pain as a (leading) symptom of mental disorders” was not included in the analysis because specific ICD-10-GM codes describing this condition do not exist. However, the topic was approached via a consideration of the severe mood disorders (without knowing if pain is a symptom) and persistent somatoform pain disorders (F45.40). In outpatient care, ICD codes with the addition G (secured) were taken into account. In inpatient care, diagnoses marked as principal, secondary, or departmental diagnoses were considered. Each insured person could have several contraindications.

Given the data structure, outpatient ICD codes could be assigned on a quarterly basis, whereas inpatient ICD codes were assigned to the discharge date on a daily basis. The OA prescription was assigned to the pharmacy dispensing date. The interruption of an OA prescription cycle was assumed if there was no OA prescription within one quarter after the second observation quarter.

For this analysis, contraindications were operationalized in temporal relation to the prescription of OAs (see example in Table 3). After the first two quarters of the observation period (selection period), the medication timeline may have been continued (prescription from observation quarter (OQ) 1–8), interrupted and resumed (see OQ4 in Table 3), or discontinued (see OQ5 to OQ8 in Table 3). For this analysis, contraindications were considered relevant only if they occurred in quarters of OA prescription. Furthermore, if a contraindication was coded in the last quarter of an OA prescription cycle, it was assumed that the occurrence of the contraindication may have led to the termination of the OA therapy and was not included as a contraindication in the analysis. If an OA was prescribed in the eighth OQ, this was considered as the last OQ of a prescription cycle, as it is unclear whether the prescription was continued afterwards.

Logistic regression was used to identify predictors for the presence of a contraindication. Two types of models were run for each contraindication group. Models A considered patients with the relevant contraindication for OA therapy vs. no relevant contraindication as dependent variable. Accordingly, these models aimed to analyze inappropriate care for individuals with an indication as well as inappropriate care and overuse for individuals without an indication (see Table 1). In addition, models B were run considering persons with a contraindication and without any indication (see Appendix A) for OA therapy vs. persons without a contraindication or persons with a contraindication, having an indication as dependent variable. Due to the absence of indications for OA therapy, these models were intended to describe overuse as well as inappropriate care only (see Table 1). Models B took a closer look at individuals presumably not having an objective justification for OA therapy, investigating whether predictors differ from those of models A, including persons with presumably objective justification for OA therapy as well.

Indications were extracted from the German guideline [1] and identified in the sample via tracer ICD codes (see Appendix A). The presence of an indication was assumed if it was coded as a secured diagnosis in the outpatient dataset or as principal, secondary, or departmental diagnoses in the inpatient dataset in the first and second quarter after inclusion of the insured in the cohort. The independent variables included in the models are listed and described in Table 4. In addition to patient characteristics such as sex and age, variables regarding the course of treatment and the utilization of medical services relevant to pain management as well as interdisciplinary therapy components were included. The latter included outpatient therapy by a specialized pain therapist, psychotherapy (EBM 35401 to 35559), psychosomatic primary health care, and an inpatient interdisciplinary-multimodal pain therapy (IMPT) (OPS 8-918.-, 8-91b.-, 8-91c.-). As geographical differences in prescription patterns have been reported for Germany [11], a variable for the insured person’s place of residence was included in the models with the characteristics east, north, west, and south. Non-guideline compliant care due to the lack of indications for the therapy was included. Due to the definition of the contraindication variable in model B, the variable “indication” could not be included in models B because of perfect correlation. Additionally, we considered whether individuals in the cohort experienced psychological and behavioral problems from opioids (ICD-10 F11.-). Since the contraindication groups “harmful use of opioids” and “mental and behavioral disorders caused by opioids” were identified via ICD-10-GM coding F11.-, the independent variable “ICD coding F11” was not included in the logistic regressions of these two contraindication groups either due to perfect correlation. The independent variables were tested for multicollinearity.

In the administrative claims data, each physician is assigned a pseudonymized physician number. Each prescription of an OA could be directly assigned to a physician. However, in outpatient care, ICD-10-GM diagnoses were recorded at the level of a treatment case. Such a treatment case could cover a time period of up to one quarter and could involve one or, in some cases, several physicians, for example, in a group practice. A physician who made a diagnosis or was involved in the same treatment case in which the diagnosis was made was assumed to have knowledge of the diagnosis. For the analysis, we investigated whether the physician’s identification number of an OA prescription corresponded with the physician’s identification numbers of the treatment cases containing contraindications. Prescription and diagnosis must have occurred in the same observation quarter. Physician’s identification numbers were applicable for outpatient diagnoses only.

Data analysis was performed using Stata 17. Significance levels were fixed at 5%. Absolute and relative frequencies of contraindications and concordant physician’s identification numbers were presented.

## 3. Results

The selected cohort included 113,476 insured persons, which accounted for 2.3% of the insureds > 17 years of the SHI company. The mean age was 71.8 years (SD 14.4); 74.6% of the included persons were female. During the follow-up period, 10% of the insured patients died. Half of the study population (49%) had a history of an OA prescription > 3 quarters in 2017 prior to inclusion. Pain therapists were consulted by 16% of the cohort. An IMPT was recorded in 3% of the included persons; psychosomatic primary care was carried out in 37% of the patients, whereas psychotherapy was used by only 3%. Further details of the cohort are presented in Table 5.

The frequencies of possible inappropriate care and, if applicable, overuse due to OA prescriptions despite contraindications is shown in Table 6. About 9% of the study population was affected by primary headaches, followed by severe mood disorders (8%) and pain in somatoform disorders (5%). The other contraindication groups occurred rather sparsely (2% and less). Applying a broader definition of the severe mood disorders (ICD-10-GM F32.- to F34.- recommended in the German guideline [1]), which also includes less severe manifestations of the diagnoses, almost half of the study population was affected by the contraindication “severe mood disorders” (44%, *n* = 50,208).

Table 6 shows the proportion of insured persons within the contraindication groups who had at least one match between the identification number of the physician prescribing the OA and making the diagnosis of a contraindication, considering prescribing physicians who may have been aware of a contraindication. A proportion of up to 79% was seen in the contraindication group of chronic inflammatory bowel disease, followed by chronic pancreatitis (72%) and primary headaches (72%). The lowest proportions were found in the contraindication groups’ harmful use of opioids (43%) and mental and behavioral disorders caused by opioids (48%).

The 16 logistic regression models presented in Table 7 show whether and to what extent certain independent variables affected the odds of OA treatment despite contraindication. Two models are presented for each of the eight contraindication groups; models A predict the presence of the respective contraindication in general, and models B predict the presence of a contraindication in the absence of an indication in the first two quarters after inclusion. In model B of the harmful use of the opioid contraindication group, the age group of those over 89 years was empty, so in this particular case, this age group was combined with the 79–89 year olds. Detailed statistical information on the 16 models can be found in the Appendix A.

The number of individuals with a contraindication in models A corresponded to the absolute numbers shown in Table 6. The study population with a contraindication in models A varies from 726 (suicidality) to 9826 (primary headaches). In models B, the number of persons was considerably smaller, ranging from 156 (suicidality) to 2080 (primary headaches). The proportion of individuals with a contraindication but no indication (models B) in relation to all individuals with a contraindication (models A) in a group varied by over 20%, with a minimum of 18% for pain in somatoform disorders and a maximum of 29% for chronic pancreatitis.

The logistic regression models showed a rather low fit (McFadden’s pseudo R^2^) of at 0.12 and lower, with a minimum of 0.02 for chronic inflammatory bowel disease (model A). Testing for multicollinearity of the independent variables showed no substantial correlations.

OA therapy, despite contraindications, was predicted by the independent variables to varying degrees. Patient-related factors showed a general trend with respect to the age group: younger age groups almost universally increased the risk for the OA prescription despite a contraindication. This was the case in patients with and without an indication (models A/models B), though the effect was more pronounced in models B (OA prescription without indication). Particularly marked were the spikes for OA prescription despite the contraindications’ mental and behavioral disorders caused by opioids without the presence of an indication (model B), with an odds ratio (OR) of 11.5 of the age group 18–49 years compared to the reference group of 70–89 years and primary headaches without the presence of an indication (model B) with an OR of 9.5. In contrast, there was no clear trend across all models for the variable sex; contraindications regarding problems with opioids and chronic pancreatitis were statistically significantly more common in men, whereas the probability for OA prescription despite other contraindications was almost universally increased in the female sex.

Regarding the patient’s medical status (history of OA prescription, indications for OA therapy, ICD F11), a longer history of OA prescription was associated with a slightly to moderately increased likelihood of the presence of a contraindication for all significant outcomes. In addition, except for chronic pancreatitis, the presence of an indication showed a slightly increased chance for a contraindication for all statistically significant results. For the six contraindications tested, the independent variable mental and behavioral disorders caused by opioids (ICD F11.-) statistically significantly increased the chances of an OA prescription in the presence of a contraindication (except for chronic inflammatory bowel disease).

The utilization of the health care service IMPT showed an OR > 1 for all statistically significant OR. The groups’ harmful use of OAs (OR = 5 and OR = 4 without indication—models B) and mental and behavioral disorders caused by opioids (OR = 4 and OR = 2 without indication—models B) should be highlighted. The utilization of specialized outpatient pain therapy, another important part of health care services for chronic pain, showed a mixed picture regarding the direction of the OR. There is an increased OR of 2 in the pain in somatoform disorders and harmful use of opioids group (model A). In general, a decrease in OR from models A to models B for this variable was observed. With regard to mental health care services, outpatient psychotherapy as well as outpatient psychosomatic primary care increased in all statistically significant cases the risk for OA prescriptions despite contraindications. Almost uniformly, the OR for these independent variables were slightly higher for models A than for models B with the contraindication of severe mood disorders showing the highest OR of 3 in model A for the outpatient psychotherapy and a rather high OR of 2 for outpatient psychosomatic primary care. The variable outpatient psychotherapy also shows a rather high OR of 3 in pain in somatoform disorders (model A).

The OR of the regional variables varied rather slightly to moderately and in part significantly with no clear trend across all contraindication groups. Here, the higher chances for chronic pancreatitis in model A in the east of Germany, an OR of 2 for suicidality in model B for the north compared with the east, and for the harmful use of opioids an OR of almost 2 in the south compared with the east stood out.

## 4. Discussion

The analysis of administrative claims data of 113,476 insured persons with the long-term use of OAs without evidence of cancer or palliative care of a German SHI comprised a predominantly female (75%) cohort of upper age (mean 72 years). Almost half of the insured already have had consistent OA prescriptions in the four quarters prior to inclusion. Prescribing OAs despite the presence of a contraindication mentioned in the German guideline LONTS [1] is considered inappropriate care and, if applicable, overuse. A descriptive evaluation of contraindications coded at the same time as the OA was prescribed showed that a higher proportion of insureds with primary headaches (9%), severe mood disorders (8%), and pain in somatoform disorders (5%) could be observed. With a consistently high proportion (43% to 79%) in the contraindication groups, the prescribing physician may have been aware of the contraindication. These proportions were lower for groups with disorders related to OA use (43% and 48%) and severe mood disorders (53%). In the logistic regressions, the different contraindication groups showed heterogeneous results. Some trends could be identified, for example, with regard to age or a longer history of OA prescriptions across all contraindication groups. The models’ fit was rather low, with the highest pseudo R^2^ in the four models concerning the harmful use of opioids and mental and behavioral disorders caused by opioids (pseudo R^2^ between 0.11 and 0.12) and model A for severe mood disorders with a pseudo R^2^ of 0.1. Overall, the rather low goodness of fit indicates that only a small part of the factors predictive for OA prescription despite contraindications could be elucidated from administrative claims data. Therefore, our analysis identified that there is OA prescriptions despite contraindication in Germany, and in many cases, this is despite possible physician knowledge of the contraindication, though only initial indications for possible predictors for this type of inappropriate care/overuse can be provided. It will be the subject of future research to detect further important predictors based on additional data sources, such as explorative qualitative research and quantitative primary data collection. Nevertheless, the OR showed stronger deflections in some areas for the independent variables so that some indications for the care of persons with CNTP could be derived.

In the severe mood disorders contraindication group, a larger proportion of insured persons appeared to be receiving inappropriate care and, if applicable, overuse according to the recommendations of the German guideline [1]. This proportion (8%) was higher than the average diagnosis prevalence of 2.2% for severe depressive disorders (ICD F32.2, F32.3, F33.2, F33.3) for the general German SHI population based on administrative claims data in 2017 [12].

Mental health problems are described consistently as a critical comorbidity for OA therapy in international guidelines, increasing the risk for abuse [13,14,15,16]. The German guideline [1] specifies the contraindication severe mood disorders with the ICD codes F32-F34. These codes also include milder manifestations of the illness, for which the Australian guideline attributes a low risk [16]. Thus, the wording of the German guideline might be imprecise with regard to the specific severity of the illness.

Looking at the utilization of health care in the contraindicated somatoform pain disorders, the OR of 2 indicates individuals with somatoform pain disorder were more likely to be treated by an outpatient specialized pain therapist. This might indicate that the diagnosis is more likely to be made in an outpatient pain therapy context; thus, the diagnosis could depend on the health care setting. Then again, difficulties in outpatient pain management in primary care due to the somatoform presentation of pain may lead to more specific pain management treatment. Younger people may be more likely to be affected by this condition in a specialized outpatient pain management context, in outpatient psychotherapy, and in psychosomatic primary care.

A diagnosis of primary headaches was documented in 9% of the cohort. Primary headaches are a common disorder in the population. If an indication for OA therapy is present, it can be assumed that the use of OAs is carefully considered in the presence of this contraindication in health care practice so that the condition should be given increased attention as a contraindication without the presence of an indication (model B).

When considering persons with OA prescription despite contraindications without an indication (models B), two basic groups of contraindications could be distinguished. In the absence of a guideline-compliant indication, it might be assumed that the diagnoses primary headaches, chronic pancreatitis, and chronic inflammatory bowel disease were more likely the decisive indications. Predictors for this type of inappropriate care as well as overuse seemed to be an especially younger age and no utilization of specialized outpatient pain therapy services. From physicians’ point of view, the other contraindications (pain in somatoform disorders, harmful use of opioids, mental and behavioral disorders caused by opioids, severe mood disorders, suicidality) do not represent a justifying diagnosis for OA therapy. Regarding the utilization of outpatient pain therapy in models A, the OR was close to or above 1. In models B, the OR for this variable was below 1 except for the contraindication somatoform pain disorders with an OR of 1.1, which is still smaller than the OR of 2 in model A for this contraindication. It may be concluded that the chances of being treated by an outpatient pain therapist despite a contraindication is lower in the absence of an indication. Furthermore, the German guideline recommends interdisciplinary therapy, which was approximated by the logistic regression models by the variable’s outpatient psychotherapy, outpatient psychosomatic primary care, IMPT, and specialized outpatient pain therapy. In the presence of these variables, the chances for OA prescription despite a contraindication in models B were predominantly lower than in models A. This may indicate a lower use of interdisciplinary therapy by persons without an indication (in the presence of a contraindication).

In regards of mental health problems caused by OAs (ICD F11), the male gender and a younger age (especially <50 years) seemed to be strong predictors. This was particularly the case in both of the models in models B. This is consistent with the risk factors for abuse and dependence, the male sex, and a younger age listed in the French guideline [15]. International guidelines consider substance misuse in general as a barrier [13] or reason for increased caution [13,15] in prescribing OAs. Persons with a longer history of OA use and those with IMPT treatment also appeared to be more likely to have this diagnosis. Thus, younger, male patients with a long-term prescription of OAs seemed to be prone to be diagnosed to have psychological problems caused by OAs.

Regarding the possible knowledge of the prescribing physician of an existing contraindication, which was evaluated by analyzing the corresponding physician numbers, it was shown that a large part of the prescribing physicians may have been aware of the disease classified as a contraindication by the German guideline when prescribing OAs. This indicates that the high number of prescriptions, despite a contraindication, was not mainly caused by an insufficient knowledge of the patients’ medical history.

Administrative claims data contain all services utilized and all diagnoses documented by the study population. All adults meeting the inclusion criteria and not fulfilling exclusion criteria could be included. This provides the possibility of minimizing the selection effects and of analyzing the health care of patients with long-term OA prescriptions under routine conditions. Therefore, external validity can be considered quite high compared to other study designs, such as RCT. Nevertheless, the survey is based on a single SHI provider. It has to be discussed whether patient characteristics and/or health care structures might differ from the general population insured by SHI. Hereinafter, the transferability of the results to the entire German SHI community (88% of the German population [17]) will be examined. The transferability of the results seems possible since the health care structures for all SHI insured are basically the same due to legal regulations. Furthermore, the data provider is the third largest SHI company in Germany in terms of the number of insured persons [18], insuring about 8% of the SHI-insured adults in Germany, and operates nationwide. On the other hand, the insured structure in the SHI “DAK-Gesundheit” is characterized by a higher age than in the SHI community (adults, 65 years and over: 36% vs. 26% [19]) and a higher share of adult women (60% vs. 53% [19]). Therefore, in view of the influence of age and sex, minor deviations in the frequency of the occurrence of the described inappropriate care/overuse are conceivable.

The increasing use of pharmaceutical opioids for non-medical purposes has become a public health threat in some countries worldwide. The World Drug Report 2022 [3] identifies a particular problem with the active ingredient tramadol for North and West Africa, the Middle East, and Southwest Asia (indications also in South Asia, Central Asia, North America, and Europe) and for the drug fentanyl in North America. From 1999 to 2020, nearly 263,000 deaths are related to the use of prescription OAs in the U.S. [20]. Overdose deaths related to prescription opioids have increased nearly fivefold from 1999 to 2020 [20]. Factors promoting higher prescription rates in the U.S. were among others identified as regions with higher rates of non-Hispanic white ethnicity, unemployment, uninsured status/Medicaid enrollment, and micropolitan status [21]. The current discussion on inappropriate care/overuse in Germany and the indications of problems detected in this analysis raise the question of whether the high level of regulation and monitoring of OA drugs led to a high feeling of security, and thus, undesirable developments have not been addressed politically. As part of the Op-US project, political recommendations will be developed. Among other things, it will be analyzed whether the current regulations are not only strict but also constructive.

Though the analysis based on administrative claims data gives important insights, it has to be noted that the data were originally collected for billing purposes. This implies certain limitations. The dataset contained diagnoses and procedures recorded by physicians. Data gaps could result from physicians’ lack of knowledge of diseases or a lack of ICD coding. Furthermore, inaccuracies in coding quality must be taken into account, such as continuous coding of a condition that no longer affects the insured person. Additionally, when considering the indications or contraindications, it was not possible to conclusively clarify the extent to which these were the causes for OA prescription or relevant in the individual case. However, since a consideration of administrative claims data allows a large cohort to be taken into account without dropouts, recall bias, or the exclusion of certain patient groups, it can be assumed that general trends in the cohort can be perceived. Conclusions should always be considered in light of the limitations of administrative claims data. With regard to the modeling of the logistic regression for the contraindication harmful use of opioids, model B had the peculiarity that the age group >89 years was empty, so for the purpose of the model calculation, this age group was combined with the 79–89 year olds, which means that the formation of the age variable is different from that of the other models.

## 5. Conclusions

Prescribing OAs despite contraindications seems to play a relevant role in the reality of care. Some predictors could be identified in the administrative claims data; future research based on additional data sources should identify causes and influencing factors. Furthermore, physicians’ reasons for deviating from the guideline recommendations should be identified so that suggestions for improved care practice can be developed.

## Figures and Tables

**Table 1 ijerph-21-00180-t001:** Definition of the terms inappropriate care and overuse based on the description of the German expert council on health and care [10] and assignment to the models A and B.

	Health Service	Improperly Provided Health Service ^b^— Contraindication Is Present
Objective Need ^a^		
yes—indication is present		inappropriate care (model A)
no—indication is not present		overuse as well as inappropriate care (model A, model B)

^a^ subjective need for OA therapy on the part of the patient is assumed for all cases. ^b^ alternatively, the health service may be “properly provided” or “not provided”.

**Table 2 ijerph-21-00180-t002:** Definition of contraindication groups and associated ICD-10-GM codes based on the German guideline on long-term opioid analgesic treatment for chronic non-tumor pain [1].

Contraindication Groups	ICD-10-GM
primary headaches	G43.-, G44.0, G44.2
pain in somatoform disorders	F45.40
chronic pancreatitis	K86.0, K86.1
chronic inflammatory bowel disease	K50.-, K51.-
harmful use of opioids	F11.1
mental and behavioral disorders caused by opioids (except harmful use—F11.1)	F11.-
severe mood disorders	F32.2, F32.3, F33.2, F33.3
suicidality	R45.8

**Table 3 ijerph-21-00180-t003:** Exemplary presentation of the consideration of contraindications in temporal relation to prescriptions of opioid analgesics. Both the occurrence of the contraindication and the three possible medication timelines are examples.

Timeline: Contraindication/Medication		Observation Quarter							
		1	2	3	4	5	6	7	8
occurrence of a contraindication (example)				x	x	x	x	x	x
medication timeline	1. continued			✓	✓	✓	✓	✓	! ^a^
	2. interrupted & resumed			!		✓	✓	!	
	3. discontinued			✓	!				

x = quarter with ICD-10-GM coding of a contraindication; orange color = quarter with prescriptions of opioid analgesics; ✓ = contraindication is considered in the analysis; ! = last quarter of a prescription cycle—contraindication is not considered; a = considered as last quarter of a prescription cycle since it is unclear whether the prescription will be continued.

**Table 4 ijerph-21-00180-t004:** Description of independent variables included into the logistic regression models.

Variable	Definition/Source	Value Specification
sex	specification in master data	- male - female
age group	specification in the master data at the time of selection, categorization into groups	- 70–89 years (reference group) - 18–49 years - 50–69 years - >89 years
region	assignment of the indication of the German federal state at the beginning of the observation period to the regions North (“Mecklenburg-Western Pomerania”, “Schleswig-Holstein”, “Hamburg”, “Lower Saxony”, “Bremen”), West (“North Rhine-Westphalia”, “Hesse”, “Rhineland-Palatinate”, “Saarland”), South (“Baden-Württemberg”, “Bavaria”), East (“Brandenburg”, “Berlin”, “Saxony-Anhalt”, “Saxony”, “Thuringia”).	- east (reference group) - north - west - south
history of OA prescriptions prior to inclusion	short- or long-term history of OA prescription: Selection period starts in the 1st quarter of 2018, inclusion criterion is OA prescription in 2 consecutive quarters. Persons included in the 1st quarter of 2018 are distinguished between persons with a long-term history (>3 quarters in 2017) and short-term history of consecutive prescriptions (≤3 quarters in 2017). Persons included after the 1st quarter of 2018 do not have a history of OA use in the quarter prior to inclusion.	- history of >3 quarters of OA prescription in 2017 and selected in the 1st quarter of 2018 - history of ≤3 quarters of OA prescription in 2017 and selected in the 1st quarter of 2018 or selected after the 1st quarter of 2018
outpatient pain therapy	at least one OA prescription from an outpatient pain therapist service in the observation period	- prescription present - prescription not present
IMPT	at least one coding of an inpatient interdisciplinary-multimodal pain therapy in the observation period	- coding present - coding not present
outpatient psychotherapy	at least one billing of a code for psychotherapeutic therapy in the observation period	- coding present - coding not present
outpatient psychosomatic primary care	at least one billing of a coding for psychosomatic primary health care in the observation period	- coding present - coding not present
psychological and behavioral disorders caused by opioids (ICD F11.-) ^a^	at least one ICD-10-GM code F11.- in the observation period	- coding present - coding not present
indication ^b^	ICD coding of an indication according to the German guideline [1] in the two quarters after selection of an individual	- indication present - indication not present

IMPT: inpatient interdisciplinary-multimodal pain therapy. ^a^ Not included in the models concerning “harmful use of opioids” and “mental and behavioral disorders caused by opioids” because there is a perfect correlation with the dependent variables due to overlapping definitions. ^b^ Not included in models B because there is a perfect correlation with the dependent variables due to overlapping definitions.

**Table 5 ijerph-21-00180-t005:** Patient characteristics of the included cohort.

Patient Characteristics	*n*/Mean ^a^	*%*/SD ^a^
sex		
female	*84,605*	*74.56*
male	*28,871*	*25.44*
age group		
18–49 years	*8571*	*7.55*
50–69 years	*35,735*	*31.49*
70–89 years	*60,136*	*52.99*
≥90 years	*9034*	*7.96*
age	71.80	14.36
region		
east	*19,022*	*16.76*
north	*26,518*	*23.37*
west	*43,316*	*38.17*
south	*24,620*	*21.70*
history of >3 quarters of OA prescription prior to inclusion	*55,684*	*49.07*
outpatient pain therapy	*17,968*	*15.85*
IMPT (inpatient)	*3545*	*3.12*
outpatient psychotherapy	*3265*	*2.88*
outpatient psychosomatic primary care	*41,998*	*37.01*
indication	*84,687*	*74.62*

^a^ Numbers and percentages are given for categorical variables and means and standard deviations for the continuous variable ‘age’. SD = standard deviation.

**Table 6 ijerph-21-00180-t006:** Number and proportion of persons with a contraindication as well as concordant physician numbers of prescriptions for an OA and the ICD coding of a contraindication.

Contraindication Group	Diagnosis of the Contraindication Group		Persons with Concordant Physician Numbers	
	*n*	%	*n*	% (of Persons in a Contraindication Group)
primary headaches	9826	8.7	7050	71.7
pain in somatoform disorders	5146	4.5	3173	61.7
chronic pancreatitis	1166	1.0	842	72.2
chronic inflammatory bowel disease	2246	2.0	1770	78.8
harmful use of opioids (F11.1)	743	0.7	322	43.3
mental and behavioral disorders caused by opioids (except harmful use—F11.1)	2309	2.0	1102	47.7
severe mood disorders	8735	7.7	4660	53.3
suicidality	726	0.6	446	61.4

**Table 7 ijerph-21-00180-t007:** Logistic regression on the presence of a contraindication.

Contraindication Group	Primary Headaches		Pain in Somatoform Disorders		Chronic Pancreatitis		Chronic Inflammatory Bowel Disease		Harmful Use of Opioids		Mental and Behavioral Disorders Caused by Opioids		Severe Mood Disorders		Suicidality	
Independent Variables	Odds Ratio															
	model A	model B	model A	model B	model A	model B	model A	model B	model A	model B	model A	model B	model A	model B	model A	model B
female	2.20 ^c^	1.83 ^c^	1.28 ^c^	1.21 ^a^	0.47 ^c^	0.45 ^c^	1.30 ^c^	1.47 ^c^	0.60 ^c^	0.50 ^c^	0.67 ^c^	0.57 ^c^	1.22 ^c^	1.16 ^b^	1.18	1.25
age group (ref. 70–89 years)																
18–49 years	3.71 ^c^	9.50 ^c^	1.80 ^c^	4.31 ^c^	1.65 ^c^	5.26 ^c^	2.77 ^c^	9.03 ^c^	5.01 ^c^	13.27 ^c^	3.86 ^c^	11.48 ^c^	1.63 ^c^	3.31 ^c^	1.50 ^b^	3.83 ^c^
50–69 years	2.35 ^c^	3.36 ^c^	1.60 ^c^	2.17 ^c^	1.73 ^c^	3.36 ^c^	1.86 ^c^	3.42 ^c^	2.10 ^c^	3.15 ^c^	1.87 ^c^	2.92 ^c^	1.54 ^c^	1.99 ^c^	1.16	1.9 ^c^
>89 years	0.39 ^c^	0.48 ^c^	0.65 ^c^	1.07	0.84	0.81	0.57 ^c^	0.70	0.43 ^a^	- ^e^	0.40 ^c^	0.51	0.78 ^c^	0.98	0.89	0.90
region (ref. east)																
north	1.00	1.09	0.83 ^c^	0.89	0.64 ^c^	0.91	0.96	1.29	1.72 ^c^	1.34	1.06	1.38 ^a^	1.13 ^b^	1.24 ^a^	1.38 ^a^	2.08 ^a^
west	1.04	1.11	0.80 ^c^	0.99	0.65 ^c^	0.92	1.11	1.66 ^c^	1.37 ^a^	1.64 ^a^	0.83 ^b^	1.17	1.47 ^c^	1.70 ^c^	1.32 ^a^	1.77
south	1.01	1.36 ^c^	1.02	1.34 ^b^	0.55 ^c^	0.95	0.95	1.65 ^b^	1.77 ^c^	1.91 ^a^	0.92	1.36 ^a^	1.36 ^c^	1.84 ^c^	1.24	1.80
history of >3 quarters of OA prescriptions prior to inclusion	1.29 ^c^	1.08	1.47 ^c^	1.48 ^c^	1.16 ^a^	0.89	1.30 ^c^	1.13	2.68 ^c^	1.99 ^c^	2.92 ^c^	2.08 ^c^	1.21 ^c^	1.06	1.17 ^a^	1.23
outpatient pain therapy	1.61 ^c^	0.82 ^b^	2.03 ^c^	1.13	1.00	0.68 ^a^	0.98	0.59 ^c^	2.07 ^c^	0.99	1.53 ^c^	0.90	1.14 ^c^	0.63 ^c^	0.97	0.44 ^b^
IMPT (inpatient)	1.38 ^c^	1.13	1.55 ^c^	1.58 ^c^	0.79	0.53	0.93	0.68	4.64 ^c^	3.67 ^c^	3.72 ^c^	2.20 ^c^	1.33 ^c^	1.13	1.02	1.30
outpatient psychotherapy	1.52 ^c^	1.19	2.22 ^c^	1.49 ^b^	0.85	0.50	1.05	1.00	0.78	0.69	0.95	0.80	3.16 ^c^	2.49 ^c^	1.08	0.58
outpatient psychosomatic primary care	1.57 ^c^	1.18 ^c^	2.61 ^c^	1.99 ^c^	1.27 ^c^	1.14	1.27 ^c^	0.98	1.92 ^c^	2.05 ^c^	2.35 ^c^	1.89 ^c^	2.23 ^c^	1.74 ^c^	2.72 ^c^	2.29 ^c^
ICD coding F11	1.14 ^b^	1.35 ^b^	1.71 ^c^	1.86 ^c^	1.84 ^c^	2.1 ^c^	1.21	1.40	- ^d^	- ^d^	- ^d^	- ^d^	2.07 ^c^	2.17 ^c^	2.14 ^c^	2.8 ^c^
indication	1.25 ^c^	- ^d^	1.34 ^c^	- ^d^	0.82 ^b^	- ^d^	1.01	- ^d^	0.97	- ^d^	0.91	- ^d^	1.16 ^c^	- ^d^	1.17	- ^d^
Pseudo R^2^	0.0834	0.0782	0.0974	0.0582	0.0331	0.0572	0.0227	0.0566	0.1237	0.1167	0.1146	0.1104	0.0678	0.0516	0.0311	0.0453
No. of persons with contraindication	9826	2080	5146	916	1166	342	2246	583	743	188	2309	593	8735	1900	726	156

IMPT: inpatient interdisciplinary-multimodal pain therapy. Model A: respective contraindication is present vs. respective contraindication is not present. Model B: respective contraindication is present and no indication for an OA therapy could be identified vs. respective contraindication is not present or respective contraindication is present an indication is present. ^a^ *p* < 0.05. ^b^ *p* < 0.01. ^c^ *p* < 0.001. ^d^ Independent variable is not included because there is a perfect correlation with the dependent variable due to overlapping definitions. ^e^ The group of >89-year-olds is empty.

## Data Availability

Data were obtained from the German Statutory Health Insurance “DAK Gesundheit” and are not publicly available due to data protection regulations.

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
