# Peer review of "Prescription of Opioid Analgesics for Chronic Non-Cancer Pain in Germany despite Contraindications: Administrative Claims Data Analysis"

_ijerph, 2024, doi:10.3390/ijerph21020180_

Round 1

Reviewer 1 Report

Comments and Suggestions for Authors

Overall

Thank you for the opportunity to read this manuscript. This manuscript provides a lot of information on what the authors were investigating.

Reading through this manuscript, there is a lot of information to the point I think it unfortunately will overwhelm readers. There was so much data collected and presented, it is hard to keep straight if the focus is specifically on HDSI intervention role expansion of if that role is merely one used to generally described pharmacy role expansion.

Abstract

Abstract does a good job introducing the project, a general insight into what prescribing looks like in Germany, and what the researchers would looking into. The methodology description does a good job at introducing generally what was done by the research team, and the results listed make readers want to look more into what this study found.

Introduction

p.2, line 42. The use of “so-called” could give a negative or judgmental connotation to some readers. It probably is not necessary for understanding.

p.2, line 60-62. This reads a little confusing to me. I think it might be better to define the project after the “Op-US project” instead of in italics between “Op-US” and “project”.

Materials and Methods

This section does a great job breaking down why specific analyses were used and what each step of the criteria would be for the study.

p.5, line 150 and line 162. Be sure to address “Error! Reference source not found.” In these spot.

Within this section, there are occasionally spots where there need to be additional punctuations (particularly commas for some sentences).

Excellent use of tables to help visualize what is being explained in this section as well.

Results

Results are presented clearly, and tables visualize information well. Tables would be able to stand alone if presented separately from the paper.

Reviewer 2 Report

Comments and Suggestions for Authors

1- The introduction provides a clear and alarming context for the opioid crisis in the U.S. and introduces the regulated prescription of opioid analgesics in Germany. It might be beneficial to elaborate on the factors contributing to the crisis in the U.S. for readers who may not be familiar with the topic.

2- Clarify whether the use of administrative claims data from a specific health insurance provider (DAK Gesundheit) might limit the generalizability of the findings. It would be helpful to discuss potential biases associated with using data from a single provider and how they might impact the external validity of the study.

    • 3-The demographic information is well-presented. Consider providing additional information about the socioeconomic status of the cohort, as this can impact healthcare utilization.
    • It would be helpful to include information on the distribution of comorbidities within the cohort, as this could potentially influence the study outcomes
    •  

    •  

Reviewer 3 Report

Comments and Suggestions for Authors

Thank you for the interesting paper.  The introduction needs to be more globally oriented and not just focused on the United States as the introduction paragraph.  Opioid crisis is a global problem as acknowledged by WHO in their August 29, 2023 Fact Sheet.  Other countries are identified including US, Canada, Guyana, Bolivia, and Dominican Republic.  US has prescription regulations and special forms for tracking opioids with some states having pretty significant restrictions with morphine equivalents as well as long-term use.  The US standards include both federal DEA and state legislative rules and regulations.  Germany is not the only country with highly regulated OA use.  

Page 2 Line 48, remove also

Line 66 Suggest reworking.  The project methods are described in more detail....

Line 67  remove that were

Line 69-73 needs rewritten as objectives and made into complete sentences.  It is very chopped up with numbers in paragraph.  Needs better flow.

Line 72 remove Furthermore

Line 100 Remove If, just start with In addition

Line 119 Remove that are

Line 131 to Line 133 remove 1. 2. and 3.  rewrite as a complete sentence or breakup into separate sentences

Line 136 remove that

Line 137 remove thus

Line 144 two types of models were run repeats itself.  reword

Line 149-150  Error message needs fixed

Line 155 remove also

Line 162 Error message needs fixed and there is an extra period there

Line 170 remove Also, start sentence with Non-guideline

Line 180 Table 4   with history of > 3 quarters --is it > 3 quarters or greater than and equal to 3 quarters, needs fixed in first and third column to be consistent

Line 180 Table 4 outpatient psychosomatic primary care, consider changing to   at least one billing of a code for ....

Line 210 Table 5  is it history of >3 quarters or >3 or equal and greater than 3

Line 222  consider   numbers of prescriptions for an OA and the ICD....

Line 233 Table 7  again >3 quarters or greater than or equal to 3 quarters

Table 7 provides OR, was there any variability with the OR (data does not show)

Line 275 consider changing to   was almost universally increased in the female sex. 

Line 282  remove Finally  start with For the six...

Line 325  Discussion about low validity ..great discussion to have about the data, but the concern here is does the low validity discount the results found, with the low validity, how do you justify conclusions?  Is it a relevant role or a potential role that requires further research?  This is significant issue that needs addressed in more detail. 

Line 342  pain disorder, 

Line 343 remove that

Line 401 dropouts as noun is one word without hyphen

Comments on the Quality of English Language

Minor adjustments

Round 2

Reviewer 3 Report

Comments and Suggestions for Authors

Thank you for your thoughtful revisions after reviewer's comments.